# ClimART: A Benchmark Dataset for Emulating Atmospheric Radiative Transfer in Weather and Climate Models

**Salva Rühling Cachay**[*]
TU Darmstadt & Mila
salvaruehling@gmail.com

**Venkatesh Ramesh**[*]
Université de Montréal & Mila
venkatesh.ramesh@umontreal.ca

**Jason N. S. Cole  Howard Barker**
Environment and Climate Change Canada
{jason.cole,howard.barker}@canada.ca

**David Rolnick**
McGill University & Mila
drolnick@cs.mcgill.ca

## Abstract

Numerical simulations of Earth's weather and climate require substantial amounts of computation. This has led to a growing interest in replacing subroutines that explicitly compute physical processes with approximate machine learning (ML) methods that are fast at inference time. Within weather and climate models, atmospheric radiative transfer (RT) calculations are especially expensive. This has made them a popular target for neural network-based emulators. However, prior work is hard to compare due to the lack of a comprehensive dataset and standardized best practices for ML benchmarking. To fill this gap, we build a large dataset, ClimART, with more than *10 million samples from present, pre-industrial, and future climate conditions*, based on the Canadian Earth System Model. ClimART poses several methodological challenges for the ML community, such as multiple out-of-distribution test sets, underlying domain physics, and a trade-off between accuracy and inference speed. We also present several novel baselines that indicate shortcomings of datasets and network architectures used in prior work.[2]

## 1 Introduction

Numerical weather prediction (NWP) models have become essential tools for numerous sectors of society. Their close relatives, global and regional climate models (GRCM) provide crucial information to policymakers and the public about Earth's changing climate and its various impacts on the biosphere. These models attempt to simulate many complicated physical processes that interact over wide ranges of space and time and seamlessly link Earth's atmosphere, ocean, land, and ice. However, due to the complexity and number of physical processes that have to be addressed, various simplifications must in practice be made, involving mathematical and numerical approximations that often have substantial statistical bias and computational cost.

One of these approximations is the *sub-grid scale parametrization* that is routinely used to approximate atmospheric radiative transfer (RT). The RT routine has traditionally represented the largest computational bottleneck in most weather and climate simulation. To speed up the computation, this routine is run only every few iterations and the results for the intermediate iterations have to be interpolated. Needless to say, this approximation of intermediate steps introduces errors in climate

---

[*]Equal contribution

[2]Download instructions, baselines, and code are available at: https://github.com/RolnickLab/climart

35th Conference on Neural Information Processing Systems (NeurIPS 2021) Track on Datasets and Benchmarks.

and weather predictions. The computational cost also means that climate models must be run at extremely coarse spatial resolutions that leave most processes unresolved and reduce the utility of information in predicting and responding to the effects of climate change.

Thanks to their better inference speed, neural network-based *surrogates* are a promising alternative to computationally slow physics parametrizations. Such hybrid modelling approaches, however, present several challenges, including accurate emulation of complex physical processes, as well as the (out-of-distribution) *generalization power* of ML models to handle environmental conditions not present in their training datasets (e.g., weather states that are not realized under current conditions).

To date, different datasets, setups, and evaluation procedures have made results in this space hard to compare. This can be attributed to the fact that no comprehensive public dataset exists, and the creation of it requires access to, and knowledge of, the relevant climate model. To address these issues and catalyze further work, we introduce a new and comprehensive dataset for the RT problem and open-source it under the Creative Commons license.

Our key contributions include:

- **ClimART: Climate Atmospheric Radiative Transfer**, is the *most comprehensive* publicly available dataset for ML emulation of weather and climate model parameterizations. It comes with *more than 10 million samples*, including three subsets of data for evaluating *out-of-distribution (OOD) generalization*.
- **Applying New Models to ClimART**. We propose multiple new models not studied in the related work, which thanks to the comprehensiveness of ClimART, allow us to *analyze the limitations of previously used models and datasets*.
- **Towards Advancing the State-of-the-Art**. ClimART's scale, unique properties, and ease of access, together with the accompanying code interface and baselines, will lower barriers for the ML community to tackle impactful challenges in climate science. ClimART also presents opportunities for spurring methodological innovation in ML via multiple *out-of-distribution test sets*, the scope for building *physics-informed ML models*, and the *accuracy versus inference speed trade-off* inherent in the problem setting.

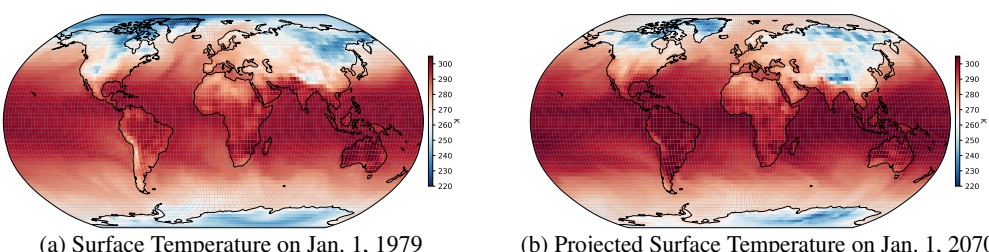

(a) Surface Temperature on Jan. 1, 1979      (b) Projected Surface Temperature on Jan. 1, 2070

Figure 1: Surface temperatures in atmospheric snapshots from 1979 and 2070.

## 2   Related Work

There have been various recent efforts to emulate sub-grid scale parameterizations by neural networks, which, thanks to their computational efficiencies, are expected to significantly speed up large-scale model simulations [6, 25, 30].

The computational burden of the RT physics motivated early pioneering work to seek out its emulation with shallow multi-layer perceptron (MLP) networks [8, 9, 15, 16], including decadal climate model simulations [17]. More recent work still focuses on using MLPs to emulate (a part of) the RT physics [19, 20, 22, 27, 28]. 2D CNNs have been also used in [19], which however treat the different input variables within the second spatial dimension instead of in the channel dimension. Prior work on such ML emulators, however, employed datasets that simplify Earth (e.g., with Aqua-planet conditions [6, 25, 30]), use a limited subset of climate model variables as predictors [15, 19, 25, 28], use manually perturbed test sets [19, 28], and generally fail to accurately probe the generalization power of ML models [19, 20, 22, 28]. The latter is particularly important, randomly-split test sets [20, 22, 28], and/or test data coming from at most two different years [19, 20, 22] can overestimate the actual skill

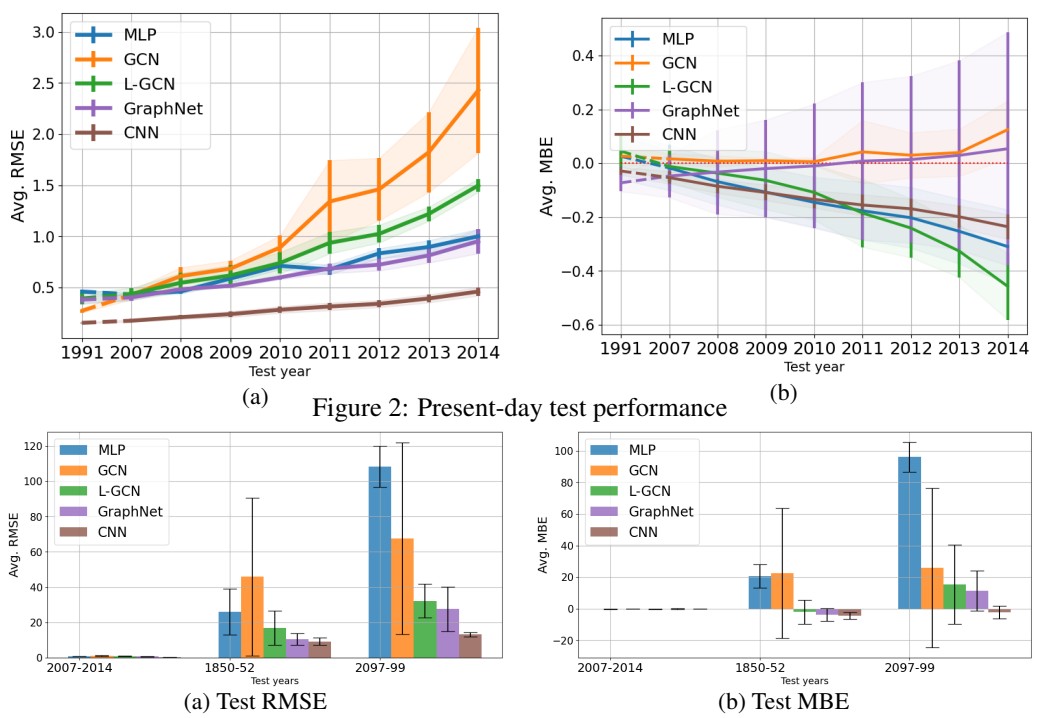

Figure 2: Present-day test performance

Figure 3: Performance as a function of the main test set year (Fig. 2) and OOD test set (Fig. 3) for our different baseline models. Metrics are in $W/m^2$ and shown as the average over the vertical and over the up- and down-welling flux errors. The leftmost x-tick for the OOD plots corresponds to the average metric of the main test years 2007-2014, for which the metrics are shown on a yearly basis in Fig. 2. Generalization to pre-industrial and future pristine-sky conditions is a particularly challenging task because of the changes in gas concentrations More structured models like a CNN or GraphNet perform significantly better than an MLP.

of the ML emulators on real world unseen data. This can be fatal when the ML emulator is to be used in long-term simulations (e.g., future climate projections).

In physics-based RT models as used in large-scale environmental models, computation of radiative flux profiles is a two-step process. The first step involves the calculation of optical properties for gases, aerosols, and clouds. The second, more computationally intensive, and arguably most erroneous step is the application of a solution of the RT equation, using the optical properties from the first step, leading to vertical profiles of radiative fluxes. Work by [27, 28] focused on a hybrid approach in which gaseous optical properties are predicted by an ML model and then passed off to a physics-based RT model for calculation of fluxes. However, [28] generates training data by just using perturbations on input variables of RFMIP[24]. [27] make a more comprehensive effort via generation of training data by combining multiple sources such as RFMIP[24], CKDMIP[12] and CAMS[1], and perturbing them to generate sufficient training data. Such perturbations, however, might lead to unrealistic input values.

The present work is closer to [19] that used a small fraction of ERA-Interim data from 1979-85 and 2015-16, Their concern was however, limited, to longwave radiative flux profiles for simplified clear-sky atmospheric conditions (without greenhouse gases like methane). Similarly to [28], they employ a test set that includes manually perturbed atmospheric states. We believe that our dataset that includes data of pre-industrial and future climate drawn from an actual climate model, is more realistic and the better choice.

## 3 Background

In the following, we introduce background and terminology that is helpful in order to better understand the problem setup and the dataset we present.

**Earth System Models**   Earth System Models (ESMs) simulate Earth's climate, including inter-actions between the atmosphere, ocean, sea-ice and carbon cycle. Within the atmosphere, the ESM computes the current state of the modelled variables which includes temperature, water vapor, aerosols, and clouds. This is done using models that discretize the Earth's atmosphere into a 3D spatial grid along latitude, longitude, and vertical dimensions, as well as a discretized timestep. At each discretized point in the horizontal, one can envision processes occurring in the vertical within a *column* that provides a *profile* of information about the state of the atmosphere, e.g., temperature. These profiles can be provided with respect to $n$ *layers* in the atmosphere, in addition to $n + 1$ *levels* (interfaces between neighboring layers), where the bottom-most level corresponds to the surface and the top-most level corresponds to the top-of-atmosphere (TOA).

**Atmospheric radiative transfer**   Radiative transfer describes the propagation of radiation. In the atmosphere, radiation transmits energy between atmospheric layers. It is classified into *shortwave* (solar), and *longwave* (thermal) radiation, which are emitted by the sun and Earth, respectively. In general, shortwave radiation is absorbed by the Earth and then re-emitted as longwave radiation; some of this longwave radiation is then re-absorbed by the atmosphere. At any given time and level of the atmosphere, *up- and down-welling fluxes* refer to the amount of radiation that is traveling up and down between the adjacent atmospheric layers. As a result of emitting and absorbing radiation, the layers of the atmosphere change temperature, which is captured in *heating rate* profiles (also called cooling rates when negative). The heating rate of any given layer can be directly computed based on the up- and down-welling fluxes of the two adjacent levels (see Appendix B.4). Ultimately, the difference in radiative flux into and out of the atmosphere is responsible for changes in the Earth's overall temperature, and the dependence of radiative fluxes on gas concentrations is the source of climate change, since human activity has increased the concentration of various greenhouse gases such as carbon dioxide, methane, and nitrous oxide.

To predict radiative flux and heating rate profiles in the atmosphere, climate scientists work with three types of models for the sky, which differ in the kinds of information they factor in: Pristine, clear, and clouds. *Pristine-sky* is the simplest, meaning that only the concentrations of gases are factored in. *Clear-sky* also includes the concentrations of aerosols, which are particles present in the air such as sulfur-containing compounds. The most general case also includes clouds.

**CanESM and its radiative transfer parameterization**   The Canadian Earth System Model (CanESM) is a comprehensive global model used to simulate Earth's past climate and the present results of climate change, as well as to make future climate projections. Figure 1 shows an example of CanESM's simulation of current (1979) and future (2070) surface temperatures. Its most recent version, CanESM5 [26], simulates the atmosphere, ocean, sea-ice, land and carbon cycle, including the coupling between each of these components. For the atmosphere it uses parameterizations to represent unresolved sub-grid scale processes like radiation, convection, aerosols, and clouds.
The radiative transfer parameterization in CanESM5, is representative of the approach used in most modern ESMs. The optical properties of a number of components are accounted for, including the surfaces, aerosols, clouds and gases (represented using a correlated $k$-distribution model). The parameterization follows an independent column approximation. This intuitively means that for a given latitude and longitude, the RT physics model takes as input only information from the 1D vertical profile of the atmospheric state at the corresponding geographical location. More details on CanESM and its RT parametrization can be found in Appendix A.

## 4   ClimART Dataset

### 4.1   Dataset collection

For our main dataset, global snapshots of the current atmospheric state were sampled from CanESM5 simulations every 205 hours from 1979 to 2014.[3] CanESM5's horizontal grid discretizes longitude into 128 columns with equal size and latitude into 64 columns using a Gaussian grid ($8192 = 128 \times 64$ columns in total). This results in $\sim$43 global snapshots per year for a total of more than 12 million columns for the period 1979-2014 and a raw dataset size of 1.5TB. Each column of atmospheric-

---

[3]The choice of 205 hours provides a manageable amount of equally spaced data while also ensuring that every hour of the day is covered since 205 is relatively prime to 24.

surface properties was then passed through CanESM5's RT physics model in order to collect the corresponding RT output: Shortwave and longwave (up- and down-welling) flux and heating rate profiles for pristine- and clear-sky conditions. The resulting NetCDF4 datasets were then processed to NumPy arrays stored in Hdf5 format (one per year), with three distinct input arrays as described in the following subsection, and one output array per potential target variable. We proceeded analogously for the pre-industrial and future climate years, 1850-52 and 2097-99 respectively (see section 4.3.1). More details are available in the Appendix B.1.

## 4.2 Dataset interface

**Inputs** We saved the exact same inputs used by the CanESM5 radiation code and augmented them by auxiliary variables such as geographical information (see Appendix B.3 for full details and description). Each input corresponds to a column of CanESM5. Its variables can be divided into three distinct types: i) layer variables, ii) level variables, iii) variables not tied to the height/vertical discretization. Examples for the two first 1D variables are pressure (occurring at both, levels and layers) and water vapour (only present at the layers). The third type of variables are comprised of optical properties of the surface, boundary conditions, and geographical information related data. We refer to this set as the *global* variables. ***The data thus has a unique structure with heterogenous data types, where 1D vertical data is complemented by non-spatial information.*** We also note that the RT problem is non-local, since the spectral composition, and thus the heating rate, at one level can depend much on attenuation, or production, of radiation at a far-removed layer (e.g., reduced absorption of solar radiation by water vapour near the surface due to the presence of reflective high-altitude cirrus clouds).

**Outputs** We provide the full radiation output of CanESM5's as a potential target for pristine- and clear-sky conditions. That is, ClimART ***comes with two levels of complexity***: pristine-sky (no aerosols and no clouds) being the simpler one compared to clear-sky, which also reflects the impacts on RT due to aerosols. It consists, for both shortwave and longwave radiation, of the up-and down-welling flux profiles and corresponding heating rate profiles (i.e. $6 = 2 \times 3$ distinct variables for each sky condition).

In our experiments we focus on pristine shortwave radiation. Our dataset, however, allows the user to choose the desired target variables based on their needs.

## 4.3 Dataset split

In the following we describe the data split that we recommend to follow for benchmarking purposes.

**Training and Validation sets** ClimART provides the complete data extracted from CanESM5, as described above, from 1979 to 2006, excluding 1991-93, as suggested data for training and validating ML models. In our experiments we used 1990, 1999, and 2003 for training, while keeping 2005 for validation.

**Main Test set** We suggest to use the data from the years 2007 to 2014 as main test set. This relatively long interval allows to test the ML model on a very diverse set of present-day conditions. To make evaluations feasible regardless of the available compute resources, we chose to subsample 15 random snapshots for each year. This results in almost 1 million testing samples.

### 4.3.1 Out-of-distribution (OOD) test sets

In order to evaluate how well a ML model generalizes beyond the present-day conditions found in the main dataset, we provide *three distinct OOD test sets* that cover an anomaly in the atmospheric state, as well as two temporal distributional shifts.

**Mount Pinatubo eruption** This test set includes conditions from the year 1991, when the Mount Pinatubo volcano erupted, and probes how well the ML model can cope with sudden atmospheric changes. The challenges arise via a sudden increase in atmospheric opacity due to high-altitude volcanic aerosol (see Appendix B.2 for more details). While CanESM5's solar RT model deals with these aerosols well, it is the specification of changes in aerosol mass and distribution (i.e., inputs to the RT model) that pose the largest challenges. Since remnants of the emitted stratospheric aerosols

remained in the atmosphere for years after the eruption, we chose to exclude the two subsequent years, 1992 and 1993, from ClimART in order to avoid data leakage during training.

**Pre-industrial and future**   This test set probes how well the ML emulator generalizes under challenging distributional shifts. For this purpose, ClimART provides historic data from the years 1850-52 and future data from the years 2097-99. In both cases, the primary challenge for ML models is that they can be expected to encounter surface-atmosphere conditions that are not present in the training dataset. The primary differences between current and *pre-industrial* conditions involve: reduced atmospheric trace (greenhouse) gas concentrations for pre-industrial times; changes in aerosol emission; and some land surface properties that arise through changes in land usage. The *future climate* data can test how well ML models extrapolate to conditions that differ from the current climatic state as a result of radiative forcing through increases in greenhouse gas concentrations. Future climatic conditions were simulated by CanESM5 based on increases in atmospheric greenhouse gas concentrations and changes in aerosol emissions that follow well-defined scenarios (see [26]) laid out for the Sixth Coupled Model Intercomparison Project (CMIP6; cf. [10]).

## 4.4   Usage

It is important to note that a ML model trained on our dataset be both *verified* and *validated* before it can be employed "operational" in a global climate or NWP model. The *verification* phase is characterized by "simple" quantitative assessment of a ML model's bias and random (conditional) errors. Once one feels that the ML model is ready to go into the dynamical model, its computation-saving aspect can be assessed against any ramifications it has on the overall forecast of the model. Ideally, a successful ML model (i.e., one that is *validated*) will simultaneously reduce computation time and have incur only statistically insignificant impacts on the overall forecast.

Furthermore, we note that the level of "success" of an ML model at estimation of radiative flux profiles can differ for weather and climate applications, for it is expected that these areas of application will tolerate bias and random errors differently. For instance, an ML model with minor, but non-negligible, bias error might be acceptable for short-range weather predictions, but could have untenable affects on longer-term climate projections. Likewise, an ML model's random errors might tend to wash-out in long, low-resolution climate simulations, but might initiate spurious extreme events in high-resolution weather forecasts.

## 4.5   Limitations

Firstly, while advances on this problem would directly benefit the whole community, the inconsistent interfaces between different climate models and their parameterizations would likely require re-training the models for those specific input-output interfaces. We note that one motivation for proposing fully convolutional and graph-based networks in our experiments, is their applicability regardless of the vertical discretization of the columns (depending on the climate model, a column might be divided into different layers). The shortcoming of MLPs, which do not enjoy this property, was also identified by [28]. Applying fully convolutional and graph-based networks to emulate parameterizations with different vertical discretization than the one trained on is an interesting direction for future work and could present a way to have ML emulators that are more generally applicable.

Secondly, our targets do not include radiation output under all-sky conditions (which, besides aerosols, includes clouds). We believe however that our otherwise comprehensive dataset will serve well as a test-bed for ML emulators under the more simple (yet complex) pristine- and clear-sky conditions. Moreover, we note that pristine- and clear-sky are routinely used in diagnostic analyses of climate and weather model results. That is, while such conditions do not often occur in the atmosphere (mostly above the troposphere), they are nevertheless computed for the entire globe in order to assess a model's cloud dynamics and compare to satellite data.

| Model | Vertical Avg. | | Surface | | TOA | |
| --- | --- | --- | --- | --- | --- | --- |
| | RMSE | MBE | RMSE | MBE | RMSE | MBE |
| MLP | $0.701 \pm 0.04$ | $-0.160 \pm 0.10$ | $0.684 \pm 0.07$ | $-0.282 \pm 0.09$ | $0.573 \pm 0.06$ | $-0.208 \pm 0.10$ |
| GCN | $1.209 \pm 0.25$ | $0.034 \pm 0.04$ | $1.260 \pm 0.70$ | $0.244 \pm 0.47$ | $0.815 \pm 0.12$ | $0.071 \pm 0.15$ |
| L-GCN | $0.878 \pm 0.09$ | $-0.179 \pm 0.10$ | $0.714 \pm 0.21$ | $-0.241 \pm 0.29$ | $0.440 \pm 0.09$ | $-0.048 \pm 0.21$ |
| GraphNet | $0.648 \pm 0.04$ | $\mathbf{-0.001 \pm 0.25}$ | $0.620 \pm 0.08$ | $\mathbf{0.017 \pm 0.29}$ | $0.434 \pm 0.08$ | $\mathbf{-0.033 \pm 0.21}$ |
| CNN | $\mathbf{0.303 \pm 0.03}$ | $-0.142 \pm 0.03$ | $\mathbf{0.265 \pm 0.03}$ | $-0.138 \pm 0.03$ | $\mathbf{0.284 \pm 0.05}$ | $-0.177 \pm 0.04$ |

Table 1: We run several neural network architectures on ClimART to emulate the shortwave down- and up-welling radiative fluxes. The reported metrics (in $W/m^2$) are averaged out over all test samples (years 2007-2014), three random seeds, as well as over the two errors for up- and down-welling fluxes. Vertical Avg. also averages the metrics over all levels (heights), see 5.1 for more.

## 5 Experiments

### 5.1 Benchmarking neural network architectures

Note that prior work usually restricted the ML model to be a multi-layer perceptron (MLP). In light of the structured data in ClimART, we aim to 1) propose more structured neural network architectures that we believe are more suitable to the task and on which we hope follow-up work can build upon; 2) study how these more structured neural network architectures compare to the unstructured MLP. Thus, we benchmark an MLP against a 1-D convolutional neural network (CNN) [18], a graph convolutional network (GCN) [14], and a graph network [4]. We now give a high-level overview over each of the architectures, which all are relatively lightweight as it is important to keep the inherent *inference speed versus accuracy trade-off* in mind. More details on it and used hyperparameters can be found in Appendix C.

- **MLP:** The MLP used for our experiments is a simple three layer MLP with the following hidden-layer dimensions: $\langle 512, 256, 256 \rangle$. As an MLP takes unstructured 1D data as input, all the input variables need to be flattened into a single vector for the MLP.

- **GCNs**, take graph-structured data as input. To map the columns to a graph, we use a straighforward line-graph structure where each node is a level or layer and is connected to the two layers or levels spatially adjacent to it above and below. To take into account the *global* information, we add it as an additional node to the graph with connections to all other nodes. The resulting graph structure, in form of an adjacency matrix, is shown in Fig.4a, where the global node has index 0, and the other nodes are spatially indexed for plotting purposes, where 1 corresponds to the TOA level and the last node corresponds to the surface. A more sophisticated graph structure is studied in section 5.2 (**L-GCN**).
  The used GCN has three layers of dimension 128.

- **Graph networks**, take graph-structured data (with node and edge features), complemented by a so-called global feature vector, as input. Thus, it is the most natural model for our task, since we can map the levels to be the nodes, layers to be the edges connecting the adjacent levels (i.e. a line-graph), and the non-spatial variables to the global feature vector. Essentially, a graph network [4] consists of multiple MLP modules, for which we use 1-layer MLPs with a hidden dimension of 128. We use a three-layered graph network.

- **1D CNN:** For the CNN model, we use a 3-layer network with kernel sizes $\langle 20, 10, 5 \rangle$ and the corresponding strides set as $\langle 2, 2, 1 \rangle$. The channels parameter is given by $\langle 200, 400, 100 \rangle$, with the last channels setting it equal to the input size. We then apply a global average over the resulting tensor to get the output. To preprocess the data for CNN, we pad the surface and layers variable to match the dimensions of levels variable. Then the result is concatenated and fed to the model.

For all the models, we use a learning rate of 2e-4 with an exponential decay learning rate scheduler and Adam [13] as optimizer. All the models are trained for 100 epochs with the mean squared error loss. We report root mean squared error (RMSE) and mean bias error (MBE) statistics over three random seeds. The results for our baseline models on pristine-sky conditions are shown in Fig. 2

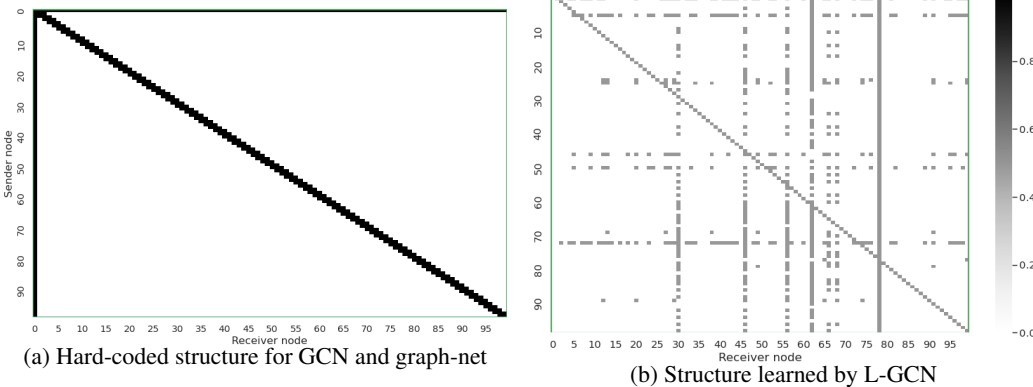

(a) Hard-coded structure for GCN and graph-net     (b) Structure learned by L-GCN

Figure 4: L-GCN, a GCN with learnable adjacency matrix, learns an edge structure very different to the diagonal structure of a line-graph (a) that is used is static adjacency matrix for the GCN and graph network baselines. This indicates that the problem benefits from *non-local information*. Notably, the many outgoing edges from *global* node (index 0, top row) indicates its importance. Node indices are sorted by height in descending order, i.e index 1 corresponds to the top-of-atmosphere node/level, and 100 to the surface level.

(more detailed in Fig. 5) and reported in Table 1. Notably all models can be seen to significantly deteriorate as a function of the test year (that becomes temporally farther away from the training years). We note that prior work could not observe this phenomenon since the test data did not cover as many years or was randomly split from the training set. We also find that the CNN architecture provides the most skillful emulation in terms of RMSE, while the GraphNet provides the least biased errors. This holds for the respective metrics computed at the TOA and surface level only, as well as when averaged out vertically over all levels. We note that the surface and TOA flux predictions are especially important, as they are directly used by the host climate or weather model. We also find that the L-GCN, which extends the GCN by a learnable edge structure module, is able to significantly outperform it, especially for TOA predictions, see next subsection.

## 5.2 Exploiting non-locality

Heating and cooling at one layer of the atmosphere depends on attenuation of radiation in all other layers. This non-locality complicates numerical simulation of the process greatly, and takes sizable amounts of computer resources to handle properly (see Appendix A for more details). Therefore, we expect ML models that can take non-locality into account to be a promising research direction. In the following we support this hypothesis with a GCN that *learns the edge structure* (i.e. the adjacency matrix of the underlying graph) as proposed in [7], denoted L-GCN. The model architecture and all other parameters are identical to the GCN. Making the connections between arbitrary layers and levels learnable relaxes the hard-coded inductive bias imposed by the highly local line-graph structure used in the standard GCN model. Indeed, not only does L-GCN outperform the GCN (Table 1), but our post-analysis also reveals that it 1) ***learns a graph structure very different to the line-graph*** used by the GCN and GraphNet (see Fig. 4), 2) gives high importance to the *global* node, which is expected given the importance of the boundary conditions and surface type of a column (see Appendix C.2 for an analysis based on the eigenvector centrality).

## 5.3 Speed

To assess the speed of our models at inference time, we speed-test the ML models for pristine-sky/clear-sky conditions on CPU and GPU, and speed-test the physics-based models for pristine-sky/clear-sky conditions for different numbers of CPUs. Note that we did not optimize the forward pass of the ML models for efficiency; thus, greater speed should be readily attainable.

For the evaluation of ML models, we make use of an instance with 4 CPUs (*2x AMD EPYC Zen 2 "Rome" 7742*), 12GB Memory and Nvidia v100 GPU. The results for different ML models for pristine-sky conditions is shown in 2 excluding the time for data loading. These results are averaged

| Model | Hardware | | |
|---|---|---|---|
| | CPU | GPU | Time (s) |
| Physics-RT | 2 | *N/A* | *3.3919* |
| | 4 | *N/A* | 3.3666 |
| | 16 | *N/A* | 2.0988 |
| | 64 | *N/A* | 1.9817 |
| MLP | 4 | × | **0.1643** |
| | 4 | ✓ | **0.0016** |
| CNN | 4 | × | 3.1870 |
| | 4 | ✓ | 0.0218 |
| GCN | 4 | × | 4.6846 |
| | 4 | ✓ | 1.6818 |
| GraphNet | 4 | × | 28.1253 |
| | 4 | ✓ | 0.1659 |

Table 2: *Pristine-sky speed benchmark* of physics-based model and ML models on different hardware configurations. The physics-based model runs serially in offline mode and is not GPU compatible yet. The ML models are all evaluated with a batch size of 8192 in CPU only and with GPU. The fastest models with and without GPU are in **boldened** and the second fastest in blue.

over 10 forward passes for an entire snapshot (8192 columns) excluding the first two warm-up passes. As expected, the MLP is fastest for both clear-sky and and pristine-sky inputs. Especially on a GPU the MLP provides, together with the CNN and GraphNet. a considerable speed-up over the RT physics. This is promising since GPUs are starting to be natively supported within the compute environments in which NWPs and GRCMs run [5], including CanESM's. When evaluated with a batch-size of 8192, the model performs **3.5x** better than with a batch-size of 512.

The physics-based RT parameterization is not GPU compatible yet so we run it by increasing the number of CPUs in the instance and average them over three runs. The RT parameterization can be significantly sped-up by increasing the number of CPUs from 4 to 16. However, the going from 16 to 64 CPUs has diminishing returns. It should be noted that this physics-based model was run in *offline* mode, where the computation is done serially. When run together with its host weather or climate model, the predictions occur in parallel.

### 5.4 OOD generalization

For evaluating the generalization of our models in OOD data, we run it on historic data (1850-1852) and future data (2097-2099). These experiments are extremely challenging given the limited size of of training set use for baseline models. Apart from this, in historic (pre-industrial) and future conditions, the values of input variables, especially those relating to the concentrations of gases vary quite a lot. For a model to be able to perform well on this data, it has to have understood the role of gas concentrations in prediction of the flux properties. As seen from the results in Fig. 3, all models degrade significantly in performance, especially for future climate conditions. However, it is notable how the models that better account for the structure of atmospheric data perform considerably better compared to MLPs: While both, the MLP and GraphNet, perform comparably well for present-day conditions with an RMSE of less than 1 $W/m^2$ (Fig. 5a), the MLP's RMSE for future conditions is above 100 while the GraphNet's stays at around 30 (Fig. 6a). Similarly, the CNN degrades "only" from less than 0.5 RMSE on the main test set, to below 18 $W/m^2$ for future-day climate conditions-

## 6 Conclusion & Future Work

We introduce a novel dataset ClimART which aims to provide a comprehensive dataset for parameterization of radiative transfer using ML models. We conduct a series of experiments to demonstrate which models are able to perform well under the inherent structure of atmospheric data in Experiments. Future work for improving upon the current baselines could include:

- Improving the model's inference speed via methods like ***weight pruning, model compression, or weight quantization***.

- Using a ***physics-informed neural network*** or loss function to predict realistic values in-line with the equations governing radiative transfer.

- ***Multi-task learning*** can be explored to emulate both shortwave and longwave fluxes or heating-rates simultaneously.

- Using Transformer-based architectures for their ability to perform well with arbitrary sequence lengths and incorporation of an ***attention mechanism.***

On the dataset side, we plan to extend ClimART to include all-sky data that includes the complexity due to clouds. We hope that ClimART will advance both fundamental ML methodology and climate science, and catalyze greater involvement of ML researchers in problems relevant to climate change.

## Acknowledgements

This research was in part supported by a Canada CIFAR AI Chair and funding has been provided by Environment and Climate Change Canada.

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
