# OpenReview forum: "ClimART: A Benchmark Dataset for Emulating Atmospheric Radiative Transfer in Weather and Climate Models"
_NeurIPS.cc/2021/Track/Datasets_and_Benchmarks/Round2 — NeurIPS 2021 Datasets and Benchmarks Track (Round 2)_

### Official Review · Reviewer_LnPt · 2021-09-16
**I am not sure about the impact to the wider community**

**Rating:** 6
**Confidence:** 2

**Strengths:**

Detailed documentation of the dataset in the appendix, detailed description of the dataset and good accessibility of the dataset. Provides a large dataset for the weather and climate model emulation community. Compared a couple of different NN architectures and provided baseline results for this dataset.

**Weaknesses:**

I have two major concerns about this work:
1. Clarity and the concise presentation of the dataset details: Although there are lots of different pieces in the generation process of this dataset using the Canadian Earth System Model, simpler language should be used to give an overview of how the data was simulated to the audience who are not well-versed in the numerical simulation in atmospheric radiative transfer community. A lot of insider jargon is used and therefore I highly doubt an average reader in the wider machine learning community would understand this manuscript easily. I have to go deep into the appendix to start gaining an understanding of what the dataset is really about.

2. Contribution and significance to the wider machine learning community: Maybe influence by the above part, the presentation of this manuscript did not show its significance to the wider ML community and how this dataset can help researchers outside this specific realm of numerical simulation of atmospheric radiation to advance their work. For the contributions summarized by the authors in Section 6, none of them strike me as a great use case for this dataset for the broader community.

A couple of other small questions/concerns:
1. Necessity of using neural emulators: Table 2 shows the speed comparison. If I understood it correctly, the Physics-RT is the numerical ground truth and it actually is the 2nd fastest method if no GPUs are used. As the authors pointed out in the caption, Physics-RT is "not GPU compatible YET", which sort of implies that this has the potential to be parallelized in the future. If that is the case, questions would raise for whether emulators are even useful since the ground truth simulations take a similar magnitude of time.

2. The out-of-distribution testing conditions:  author provided three different out-of-domain testing conditions. I am not sure what the implications are for models to behave differently on these test sets given that regularizations are not

3. In Figure 3 the error distribution is not visible as they overlap with each other. Also if there is no temporal nor logical reason (for x-axis) for Figure 3 to use a connected line plot. Maybe consider using a bar plot for that?


**Additional Feedback:**

*Post dicussion period

First of all, thanks for the crafted rebuttal in such a short time! Sorry for the part-missing question as I didn't check thoroughly the content.

After viewing the author's rebuttal (and modified version), I am happy to change my review score to "6: Marginally above acceptance threshold" as they addressed my major concern of necessity of ML emulators. Although I still think improving the simulation code (to make it GPU-friendly) is a more promising goal, as authors pointed out, this dataset is still valuable before that actually happens.



**Clarity:**

As noted in the weaknesses sections, the clarity of this manuscript in the main paper needs some significant improvement, a general ML audience who is not well-versed in the numerical simulation in atmospheric radiation.

**Correctness:**

I belive the claims made in the submissions are generally correct, the dataset construction is through large simulation physics-based model and are sound.

**Documentation:**

Documentation for the underlying physics simulation is written but it lacks connections to the general ML community to understand. The data is available and public and well maintained.

**Ethics:**

No ethical concerns has been found in this manuscript.

**Relation To Prior Work:**

The relationship to prior work is properly discussed. However, some more specific and concrete descriptions of the other publicly available datasets are desired, instead of a single sentence listed their existence and their drawbacks compared to this work.

**Summary And Contributions:**

The paper proposed a large (10M) dataset that addresses the numerical simulations emulation using Machine Learning dealing with the atmospheric radiative transfer calculation. The documentation of the dataset seems very complete and detailed. Provided three subsets of data for the out-of-distribution generalization comparison.

---

> ### Author Response · Authors · 2021-09-29
> **Response to Reviewer LnPt**
>
> We would like to thank the reviewer very much for these thoughtful and detailed questions as well as feedback. We respond to them below.
> ## Major concerns
> ### Clarity:
> We have uploaded a draft in which an additional summary on radiative transfer and background on frequently occurring climate jargon has been added to the background section, intended for an audience without any climate science expertise. We have still included various technical language in the full sections, as we believe this is important for any ML practitioner wishing to work on this dataset - this language provides pointers for understanding the context behind the dataset. We would appreciate any specific feedback you can give on areas we can further improve in this regard.
> ### Significance to the wider community:
> We believe the significance of the dataset is twofold: First, we provide a case that will be of interest to a general machine-learning audience interested in out-of-distribution generalization, physics-based tasks, and a tradeoff between speed and accuracy. For example, we include data under future atmospheric conditions that reflect climate change, representing a significant distributional shift. Distributional shift is a very significant problem for the wider ML community and climate change has often been cited as a key example of this - without a relevant dataset, this is difficult for the machine learning community to effectively engage with.
>
> Second, there is a significant component of the NeurIPS community that is interested in ML innovations for climate specifically. Last year, for example, the climate change workshop at NeurIPS included 2,000 attendees and 100 posters. It is by this audience that our paper is most urgently needed - ML researchers who are also somewhat literate in climate science, and for whom this dataset fills an important gap.
> ## Other questions and concerns
> ### Necessity of ML emulators:
> A physics RT-model that can take advantage of GPUs could indeed result in significant speed-ups. *However,* it is unclear how feasible this is, given that the physics parameterization is not easy to port to a GPU with good performance. In light of this uncertainty, we feel that it is very reasonable to work on ML emulators of which we know that they can be fast at inference, especially since GPUs are a key ingredient for next generation weather and climate models (see e.g [1, 2]). We agree with [1] (added to our references in the revised version) in that *“As tests to use machine learning accelerators within Earth-system models are in their infancy, the weather and climate community is largely unprepared to use hardware optimized for machine learning applications. On the other hand, the use of machine learning accelerators and low numerical precision comes naturally when using deep-learning solutions within the prediction workflow, in particular if used to emulate and replace expensive model components that would otherwise be very difficult to port to an accelerator, such as the physical parameterization schemes or tangent linear models in data assimilation. Thus, machine learning, and in particular deep learning, also shows the potential to act as a shortcut to HPC efficient code and performance portability.”*
>
> [1] Bauer, P., Dueben, P.D., Hoefler, T. et al. The digital revolution of Earth-system science. Nat Comput Sci 1, 104–113 (2021). https://doi.org/10.1038/s43588-021-00023-0
>
> [2] https://blogs.nvidia.com/blog/2021/06/29/canadian-weather-forecasts-infiniband-networking/
> ### Out-of-distribution sets:
> The question seems to be missing a part, and we are not sure what regularization is referring to in this context. Would it be possible to clarify?
> ### Figure 3:
> Thanks for the feedback! In the revised draft, we are using bar plots, as suggested by the reviewer (also for Fig. 6, which is the more complete version of Fig 3). Indeed, this change enhanced the visibility of the error distributions.

---

> > ### Comment · Reviewer_LnPt · 2021-10-04
> > **Updated score**
> >
> > First of all, thanks for the crafted rebuttal in such a short time! Sorry for the part-missing question as I didn't check thoroughly the content.
> >
> > After viewing the author's rebuttal (and modified version), I am happy to change my review score to "6: Marginally above acceptance threshold" as they addressed my major concern of necessity of ML emulators. Although I still think improving the simulation code (to make it GPU-friendly) is a more promising goal, as authors pointed out, this dataset is still valuable before that actually happens.

---

### Official Review · Reviewer_r8dX · 2021-09-20
**A good climate dataset**

**Rating:** 7
**Confidence:** 3
**Correctness:** Good
**Clarity:** Good

**Strengths:**

Since both climate change and weather prediction are critical problems facing our society, I think the authors studied a very important issue. It will have a great research impact if we can use approximate machine learning (ML) to achieve similar performance as explicitly compute physical processes. A good dataset is very important for this kind of research.

As mentioned by the authors, this dataset also has some interesting properties such as multiple out-of-distribution test sets, underlying domain physics, and a trade-off between accuracy and inference speed.


**Weaknesses:**

As mentioned by the authors, their targets do not include radiation output under all-sky conditions (which, besides aerosols, includes clouds). That should be improved. Another problem is how to build consistent interfaces between different climate models and their parameterizations.

**Additional Feedback:**

N/A

**Documentation:**

Good

**Relation To Prior Work:**

Good

**Summary And Contributions:**

In this paper, the authors made a new weather/climate dataset by the name of ClimART.
ClimART is a super-large dataset, which has over 10M samples (present, pre-industrial, and future climate conditions). ClimART is based on the Canadian Earth System Model.

Since both climate change and weather prediction are critical problems facing our society, I think the authors studied a very important issue. It will have a great research impact if we can use approximate machine learning (ML) to achieve similar performance as explicitly compute physical processes. A good dataset is very important for this kind of research.

As mentioned by the authors, this dataset also has some interesting properties such as multiple out-of-distribution test sets, underlying domain physics, and a trade-off between accuracy and inference speed.

As mentioned by the authors, their targets do not include radiation output under all-sky conditions (which, besides aerosols, includes clouds). That should be improved. Another problem is how to build consistent interfaces between different climate models and their parameterizations.

---

> ### Author Response · Authors · 2021-09-27
> **Response to Reviewer r8dX**
>
> We would like to thank the reviewer very much for their thoughtful feedback. We are happy that you see the value of our comprehensive ClimART dataset.
> ### All-sky conditions
> We are actively working on including and running exploratory baselines on all-sky conditions in the next iteration of the dataset. This will be an extension to the existing ClimART dataset, and the current data will, of course, remain as-is. It is important to note that the atmospheric conditions included in ClimART already are commonly used as a difficult test case for atmospheric physics models and ML emulators, and are also directly used frequently in climate simulation. The complexity brought on by the aerosols due to reflection, scattering and transmission of various particles which interact with them makes it a tough test case.
> ### Inconsistent Climate model <-> RT-physics parameterization interfaces
> Unfortunately this is an issue over which we have little control. The main differences are the set of input variables and the vertical discretization (spatial dimension). For the first, there will be no alternative other than re-training the ML model. The latter has in fact been a motivation for us to propose ML models that can be applied regardless of the spatial dimensions (i.e. fully convolutional networks, and GNNs). Applying them to emulate parameterizations with different vertical dimensionality than the one trained on would be very interesting and could present a way to have ML emulators that are more generally applicable.

---

### Official Review · Reviewer_L8AC · 2021-09-22
**Interesting Dataset on an Important Real-World Challenge**

**Rating:** 7
**Confidence:** 3
**Clarity:** The paper is very clearly written.

**Strengths:**

- The domain of the data is highly relevant in the current era of climate changes and global warming.
- The authors have done a good job of providing out-of-distribution test sets for testing the robustness of ML regression techniques.
- The paper is very nicely written.

**Weaknesses:**

- The documentation is suboptimal and the only available link is to a Google Drive file with a shell script that downloads the data. Thorough documentation, README, tutorials, etc., are naturally expected at this level.

**Additional Feedback:**

N/A

**Correctness:**

- The claims are mostly correct. I cannot judge whether the task of the dataset is really the most important challenge for climate science, but my assessment is that the problem is a relevant one.

**Documentation:**

The documentation is poor and a major effort needs to be spent on improving the documentation of the dataset.

**Ethics:**

There are no ethical concerns I could spot.

**Relation To Prior Work:**

Discussed clearly.

**Summary And Contributions:**

This paper proposes a new dataset for estimating the atmospheric radiative transfer (RT) with over 10 million samples from a collection of climate conditions spanning over 40 years of duration. The authors divided the dataset into training, validation and test sets and included out-of-distribution test sets to cover special cases: an anomaly (mount Pinatubo eruption), future climate (based on simulations), pre-industrial (1850-1852). In addition, the paper demonstrates the performance of a set of typical manifold regression baselines and discusses their performance.

---

> ### Author Response · Authors · 2021-09-28
> **Response to Reviewer L8AC**
>
> Thank you for your thoughtful comments! We agree that ClimART covers a problem very relevant to climate science.
> ### Documentation:
> We have open-sourced the code now at this link: https://github.com/RolnickLab/climart (this will also be added to the revised paper that we will upload shortly):
> This includes the code, README with instructions for dataset download and reproduciblity, as well as high-level documentation of training parameters and baseline experiments. Since we also want the community to easily use our dataset, we will document it more extensively within the repository (and include our documentation from Appendix B, C), clean the code more, and add Jupyter notebooks as tutorials.

---

### Official Review · Reviewer_grPM · 2021-09-23
**Emergency Review**

**Rating:** 6
**Confidence:** 2

**Strengths:**

Preface: As an emergency reviewer, I have to say this field (i.e. climate/weather) is well outside my domain expertise.

1. *Significance*: To the best of my knowledge subsequent to a brief literature review, it appears that the authors' contributions are as stated---that is, there is indeed an apparent need for standardization across multiple threads of research to better gauge progress and compare different approaches to deploying machine learning models within RT and weather/climate modeling more broadly. This appears in line with several recent such "call-to-arms" works in medicine and reinforcement learning, where having any amount of such standardization is much better than individual works picking their own convenient settings for study.

2. *OOD Test Sets*: Another strength is the presence of explicitly identified OOD test sets, which appear selected to handle (1) distribution shift in atmospheric opacity, (2) forward distribution shift in time, and (3) backward distribution shift in time.

**Weaknesses:**

Two questions:

1. *Benchmark Models*: Are there not more interesting models that have been proposed, than the described vanilla MLPs, GCNs, graph networks, and 1D CNNs? In the related work, has anything else been done beyond these, that can also be benchmarked?

2. *Downstream Use*: RMSE and MBE are used. Could the authors comment on how these models are typically used (i.e. is the typical scenario that one or two specific endpoints need to be predicted using a pre-defined metric, or would the models need to "simulate" a wider range of variables for multiple purposes)?

**Additional Feedback:**

Please provide some clarity into the above questions.

**Clarity:**

The paper is quite well-written; I had no issues with understanding the aims, procedures, contributions, and results.

**Correctness:**

As an emergency reviewer, the background knowledge (i.e. climate/weather) required to issue a confident assessment of this lies outside my domain expertise.

**Documentation:**

These are extensively documented in Appendices B and C.

**Relation To Prior Work:**

To the best of my judgment, the Related Work section appears to span an appropriate range of relatively recent work in this area.

**Summary And Contributions:**

This paper presents a large dataset that facilitates comparisons between research works that propose different machine learning models to replace subroutines in numerical simulations of weather and climate models (specifically, radiative transfer calculations). The goal is to standardize benchmarking across different techniques, which have hitherto been evaluated based on different datasets, setups, and metrics. Secondarily, the authors propose new models not studied in related work, as well as limitations of previously used models and datasets. Thirdly, the authors also identify multiple out-of-distribution test sets, the possibility of building physics-informed models, and the challenge of trading off accuracy versus inference speed.

---

> ### Author Response · Authors · 2021-09-28
> **Response to Reviewer grPM**
>
> We would like to thank the reviewer for their useful feedback and especially for taking the time as an emergency reviewer! We respond below to the helpful comments and questions:
> ### Question 1 on Benchmark Models:
> All the related work that we are aware of (to which we point to in the related work) has only used MLPs. [15] also explored simple 2D CNNs where the different variable types are treated in the second spatial dimension as opposed to the feature/channel dimension in our 1D CNNs (first spatial dimension is the height).
> ### Question 2 on Downstream Use:
> As mentioned in section 4.4, on the usage of ClimART, the importance of the metrics will depend on the host climate or weather model. For climate/long-term simulations, low bias errors (MBE) are particularly important, while weather/short-term simulations will mainly require a low RMSE. In the optimal case, errors are low for both metrics, which is definitely possible -- it is just important to report both metrics, we feel.
> Unfortunately, we do not understand the questions in brackets, and what ‘endpoints’ is referring to. Could you please explain that?

---

### Author Response · Authors · 2021-09-29
**General Response**

We would like to thank all reviewers again for their feedback, effort, and time spent reviewing our draft.
We have *uploaded a revised version* that we feel improves the draft, in response to certain points raised by the reviewers.
**All significant changes are highlighted in purple.**

The changes include:
- We have open-sourced our preliminary code, dataset interface, and documentation (in addition to the documentation in the Appendix) at the following URL (added to the revised draft as a footnote): https://github.com/RolnickLab/climart
- A new background paragraph on atmospheric radiative transfer (lines 102-120) that we hope will increase the clarity of the draft, and make the manuscript accessible to a larger audience by introducing domain-specific jargon explicitly. This is complemented by minor revision of some other sentences that we hope will improve clarity.
- Less recent prior works that pioneered this field, as early as 1996, by using shallow MLPs have been added to the related work section.
- To emphasize the significance of the three out-of-distribution test sets, and increase the clarity of that section, we have moved parts of the corresponding appendix into the main text, and added a few new sentences.
- Figure 3 and 6 have been transformed to bar plots in order to increase the visibility, as suggested by Reviewer LnPt
- Inspired by the weakness that Reviewer r8dx points out, we added a short discussion to the Limitations section on how our newly proposed baselines can help to attain emulators that are more generally applicable (regardless of the vertical discretization of the atmosphere in the host climate or weather model).

---

### Decision · Program_Chairs · 2021-10-10

**Decision:**

Accept

**Comment:**

This paper describes a large scale (>10 mln) dataset of atmospheric radiative transfer data spanning a large time period (40 years) and has various out of distribution test conditions. It also provides a good set of benchmark results. The reviewers agree that this dataset is an important contribution to enable the development of ML methods that help tackle some of the problems with climate change and weather. The paper is well written, and the dataset is well documented. For this reason I recommend acceptance,